# Device Development for Detecting Thumb Opposition Impairment Using Carbon Nanotube-Based Strain Sensors

**DOI:** 10.3390/s20143998

**Published:** 2020-07-18

**Authors:** Tomoyuki Kuroiwa, Akimoto Nimura, Yu Takahashi, Toru Sasaki, Takafumi Koyama, Atsushi Okawa, Koji Fujita

**Affiliations:** 1Department of Orthopaedic and Spinal Surgery, Graduate School of Medical and Dental Sciences, Tokyo Medical and Dental University, Tokyo 113-8519, Japan; kuroiwa.orth@tmd.ac.jp (T.K.); t-sasaki.orth@tmd.ac.jp (T.S.); koya.orth@tmd.ac.jp (T.K.); okawa.orth@tmd.ac.jp (A.O.); 2Department of Functional Joint Anatomy, Graduate School of Medical and Dental Sciences, Tokyo Medical and Dental University, Tokyo 113-8519, Japan; nimura.orj@tmd.ac.jp; 3AI Group, Department of 1st Research and Development, Yamaha Corporation, Shizuoka 430-0904, Japan; yu.takahashi@music.yamaha.com

**Keywords:** motion analysis, thumb opposition, carbon nanotube sensor, carpal tunnel syndrome, device development, diagnostic device

## Abstract

Research into hand-sensing is the focus of various fields, such as medical engineering and ergonomics. The thumb is essential in these studies, as there is great value in assessing its opposition function. However, evaluation methods in the medical field, such as physical examination and computed tomography, and existing sensing methods in the ergonomics field have various shortcomings. Therefore, we conducted a comparative study using a carbon nanotube-based strain sensor to assess whether opposition movement and opposition impairment can be detected in 20 hands of volunteers and 14 hands of patients with carpal tunnel syndrome while avoiding existing shortcomings. We assembled a measurement device with two sensors and attached it to the dorsal skin of the first carpometacarpal joint. We measured sensor expansion and calculated the correlation coefficient during thumb motion. The average correlation coefficient significantly increased in the patient group, and intrarater and interrater reliability were good. Thus, the device accurately detected thumb opposition impairment due to carpal tunnel syndrome, with superior sensitivity and specificity relative to conventional manual inspection, and may also detect opposition impairment due to various diseases. Additionally, in the future, it could be used as an easy, affordable, and accurate sensor in sensor gloves.

## 1. Introduction

In recent years, biomechanics research using digital devices has been very active and made many reports [1,2,3,4,5,6]. Among them, hand- and finger-sensing research is in demand in various fields, such as augmented reality, robotic hands, and rehabilitation, and is widely used [4,7,8].

In the study of the hand, the thumb is essential, as it performs 50% of hand functions, to which its greatest contribution is opposition [9,10,11]. Opposition is the movement of the thumb facing the other fingers, which includes the three elements of abduction, pronation, and flexion movements, all of which are indispensable [12,13,14]. However, most of the sensing studies on the thumb measure only abduction and flexion motions [2,3], and some methods, which can also evaluate pronation motion, are generally very complicated and require a large apparatus [15,16,17]; thus, sensing of thumb opposition motion for daily use has not yet been established.

Even in the field of the practice of hand surgery, there are many diseases in which opposition movement is impaired, and evaluation methods to detect them are important. Traditionally, the Kapandji score [18], which is an evaluation method for the function of the thumb, has been used as an evaluation method for opposition impairment [19,20,21]; however, this evaluation method has some shortcomings; for example, no significant difference was found between patients with opposition impairment and healthy subjects [22,23,24]. Thus, it has been found that it is not an entirely appropriate method for evaluating opposition motion. There are also reports of attempts to accurately assess opposition motions, such as computed tomography [17,25] and optic motion capture [26], but these are complicated and cause radiation exposure, making them almost impossible to use in daily practice. Therefore, to compensate for these shortcomings, a method that can evaluate opposition movements with continuous variables, can be easily performed, and is not invasive is important. Previously, we have reported a method for measuring thumb pronation in patients with severe carpal tunnel syndrome (CTS) using a gyroscope [27]. Severe CTS causes muscle weakness with atrophy of the thenar muscles and, consequently, impairs thumb palmar abduction and pronation movement. As a result, although a decrease in these angles could be detected, the method could not overcome the limitation that it can be affected by skin stretching, which is an inherent problem with wearable sensors.

Hence, in this study, we focused on a carbon nanotube (CNT)-based strain sensor, which can evaluate expansion length based on changes in electric resistance [28]. This device is small, thin, and accurate; thus, it can be used for analysis with almost no effect on motion even when it is attached [29]. Consequently, motion capture using the sensor has been often reported in the field of biomechanics and robotics in recent years [28,30,31,32]. Moreover, as it is soft and elastic, it can be applied on curved surfaces and movable parts with skin stretching [32,33]. In addition, it is cheap and easy to handle; therefore, it is expected to be used in evaluating movement in the living environment and in the field of virtual reality in the future. Some reports on motion analysis using stretchable sensors have reported the ability of these sensors to evaluate not only flexion and extension but also rotation through its inverse correlation with their attachment at different angles [34] as well as to evaluate the angles of the upper extremity joints accurately [35]. These findings suggest that the strain sensor can be an economical and easy method for evaluating thumb opposition.

Thus, we hypothesized that the CNT strain sensor can be used to detect thumb opposition impairment in patients with severe CTS. To verify this hypothesis, we first developed an evaluation device combining CNT strain sensors and evaluated its ability to detect thumb opposition movement in healthy people. Second, we conducted a comparative study between volunteers (controls) and persons with CTS and evaluated whether the device could detect thumb opposition impairment.

## 2. Materials and Methods

This cross-sectional study was performed according to the Strengthening the Reporting of Observational Studies in Epidemiology guidelines [36] and approved by our institution’s institutional review board. The study protocol conformed to the ethical guidelines of the 1975 Declaration of Helsinki. All participants signed a written informed consent.

### 2.1. Participants

Between August 2018 and February 2019, we included 20 hands of 11 volunteers as a control group and 14 hands of 13 patients with preoperative CTS before surgery as the CTS group. The chief complaint and trauma history of their hands of the participants was obtained at recruitment. We acquired the medical history of the patients with a medical interview and physical findings from the induction tests of CTS. The X-ray images of patients were evaluated to determine if there are bony deformities or calcifications in the carpal tunnel.

Patients diagnosed with primary CTS and who were planning to perform carpal tunnel release were included in the CTS group. We diagnosed primary CTS on the basis of the following criteria: (1) finger numbness; (2) positive physical findings of CTS, such as Phalen’s test and Tinel-like signs; and (3) abnormal results of a nerve conduction velocity (NCV) test, based on the Padua’s classification [37]. The exclusion criteria were as follows: (1) a history of hand injury, surgery or recurrence after the release of the carpal tunnel; (2) positive physical findings and imaging findings of osteoarthritis on the first carpometacarpal (CM) or thumb metacarpophalangeal joints, which can potentially affect thumb motion; (3) a suspicion of cervical spine disease; (4) a space-occupying lesion in the carpal tunnel, observed using magnetic resonance imaging, which could compress the median nerve; or (5) higher than mild NCV values, which may not yet have impaired the thumb movements.

We included volunteers as the control group who had performed a total hip replacement in our hospital and whose age and sex matched with those of patients in the CTS group. The exclusion criteria were as follows: (1) a history of hand injury, a hand that underwent surgery, thumb pain, or finger numbness; (2) positive physical findings of CTS; and (3) those with suspicion of osteoarthritis of the thumb metacarpophalangeal or first CM observed on X-ray imaging. The reason for recruiting patients who underwent total hip arthroplasty in the control group was that these patients performed routine radiography of the hand preoperatively to assess the effect on the use of T cane in our hospital; therefore, additional radiation exposure was unnecessary in such patients.

In an additional experiment for accuracy verification, four examiners used the same method on five hands of five healthy volunteers. Previous reports were referred for the number of raters, and accordingly, the number was set to be equal to or to exceed the number included in these reports [38,39,40].

### 2.2. Physical Examination and NCV Testing

We collected the following data before the measurement in this study. All physical findings, including the Kapandji score, were taken by experienced hand surgeons through a physical examination. The extent of the thenar muscle atrophy was evaluated in four stages through visual examination [41]. All NCV tests were performed and evaluated by experienced neurologists. In the additional experiment, none of the hands had a history of injury or disease.

### 2.3. Apparatus

We assembled a device using two CNT sensors (Yamaha Corporation, Shizuoka, Japan) [42]. In this sensor, 1 millimeter-long multiwalled CNTs were unidirectionally aligned and sandwiched between elastomer layers and the elastomer was synthesized urethane resin, which exhibits low elasticity and an affinity for human skin. The sensor could measure its extension and contraction as an electric voltage change. It could be stretched by up to 200% of the length and could exhibit a short sensing delay of less than 15 ms. In addition, it could withstand 180,000 cycles of expansion and contraction of up to 1.3 times of the length. We fixed the two CNT sensors (length: 17.5 mm, width: 1 mm) in an intersecting fashion at 90° to the stretch synthetic leather (thickness: 0.5 mm, Young’s modulus: 8 MPa) and the sensors connected to a conducting wire that was made of silver-plated nylon fiber. Moreover, sensors and connecting wires were placed on the leather by covering with urethane film (thickness: 0.05 mm, Young’s modulus: 20 MPa) (Figure 1). 

The device was connected to a laptop or personal computer (HP ProBook 450 G2; Hewlett-Packard, San Jose, CA, USA) that logged the sensor stretching (sampling rate: 200 Hz). The software SyncRecordT (ATR-Promotions, Kyoto, Japan) was used for the analysis. Because this device was a wired connection, it could measure continuously for about 40 h.

### 2.4. Measurements

The device was placed on the dorsal side of the thumb CM joint so that the two sensors were inclined at 45° with respect to the joint surface. On the device, a stretchable vinyl tape, which was molded specially for the measurement, was attached to fix the device to the skin (Figure 2).

We instructed the participants to first move their thumb from the adduction position to the palmar abduction position and then return to the adduction position. We regarded this as the sequence of actions and instructed the participants to perform this sequence 10 times. During the motion, a metronome was sounded once every 0.5 s, and we instructed the participants to move their thumb position whenever they could hear the sound; furthermore, the examiner maintained the participants’ wrists and second metacarpal joints in the stationary position.

### 2.5. Analysis

We calculated the confidence interval for the ratio of female participants between both groups. We defined the sensor from the proximal ulnar side to the distal radius as Sensor 1 (orange line in Figure 3) and the sensor from the proximal radial side to the distal ulnar side as Sensor 2 (blue broken line in Figure 3). Using expansion and contraction values, which were translated from the measured voltage, we calculated the average correlation coefficient (ACC) of the expansions and contractions of Sensors 1 and 2 during the motion. The calculation was performed using a moving average (Figure 4 and Figure 5) and after excluding the values of ACC at 1 s from the start and end, as the thumb was not moving in all the experiments at these time points (see the beginning and end of graphs in Figure 4 and Figure 5). Data on age and ACC are presented as median with an interquartile range. The Mann–Whitney U-test was used to compare the differences between the groups. To evaluate the utility of this method and the Kapandji score, the receiver operating characteristic (ROC) curve was plotted using the mean correlation coefficient or score, and the area under the curve (AUC) was calculated. A power analysis was performed based on the ACC. To evaluate intrarater reliability, the average intraclass correlation coefficient (ICC) (1, *k*) was evaluated in each of the four raters, and to evaluate interrater reliability, the ICC (2, *k*) was calculated in the four raters in an additional experiment.

A *p*-value <0.05 was considered statistically significant. We performed a power analysis based on the pronation angles. The sample size needed to detect a 0.5 difference in the correlation coefficient of the two groups, presuming an overall standard deviation of 0.4, with 80% power, was estimated to be 11 participants per group.

## 3. Results

### 3.1. Patient Characteristics

Demographic features are presented in Table 1. The 95% confidence intervals of the number of females and of the hands with cane ratios between both groups were −0.237 to 0.209 and −0.273 to 0.329, respectively. The features of patients with CTS are shown in Table 2. The Kapandji score of participants are shown in Table 3.

### 3.2. Measurement Data

The mean ACC was −0.58 (−0.82 to −0.35) and 0.04 (−0.41 to 0.60) in the control and CTS groups, respectively, with a significant difference (*P* = 0.023, Figure 6). The average ICC (1, *k*) (for intrarater) was 0.80, and the ICC (2, *k*) (for interrater) value was 0.76 in the additional experiment. The ROC curves with the device and the Kapandji score are presented in Figure 7. Their AUC values were 0.73 and 0.36, respectively. If the threshold was set to −0.25 for the device and 0.95 for the Kapandji score, the sensitivity was 0.71 and 0.36, respectively, and the specificity was 0.8 and 0.9, respectively.

## 4. Discussion

Using a device combining two CNT sensors, we evaluated the correlation of expansion and contraction of sensors during thumb palmar abduction movement. A moderate negative correlation was recognized in the control group, but in the CTS group, ACC was significantly higher and the correlation was lost.

We have previously successfully measured thumb opposition movements non-invasively using a gyroscope [24,27]. The previous method had limitations: the device could interfere with the thumb movement and be adversely affected by skin stretching. In this study, we focused on the CNT sensor, which is thin and elastic enough to fit the skin stretching, and easy to adapt to the joint’s shape. A previous report showed an inverse correlation between the expansion and contraction of two stretchable sensors during lumbar rotation by attaching the sensors obliquely [34]. The same results were obtained in the present study; thus, we can say that the assembled device could evaluate the rotation. In addition, although there have been reports on wrist rotation, flexion, and extension that were evaluated using stretchable sensors, to the best of our knowledge, this is the first study to evaluate thumb pronation using a stretchable sensor.

Furthermore, this report, as mentioned above, showed that the expansion and contraction of the two sensors were synchronized during flexion [34]. In our study, the ACC was significantly higher and the correlation between the two sensors was lost in the CTS group. It can be thought that a higher ACC indicates a deterioration of the pronation because, as mentioned above, the more the thumb flexes, the more these sensors correlate, and the more the thumb rotates, the more is the correlation inverse. Thus, the difference in the results between the groups suggested that flexion became fairly dominant over rotation during thumb opposition movement in the CTS group. It is known that severe CTS impairs thumb pronation during opposition movement [14,43,44]; therefore, these results are considered clinically plausible.

The AUC value indicated that the method had moderate accuracy [45,46], and the values of sensitivity and specificity were superior to those of the evaluation of opposition impairment with manual inspection that has been reported previously [47]. Moreover, in this study, the sensitivity of the Kapandji score was low, as previously reported [23], and our method was superior. However, the specificity of the Kapandji score was higher than ours; thus, we suggest that these methods are better used together. Furthermore, the ICC value indicated excellent intrarater and interrater reliability of the method [48]. These results suggest that the device was valuable and sufficiently accurate for clinical practice. Moreover, it is economical, wearable, and can be used non-invasively without the use of any special equipment. Thus, it may be applicable as a new evaluation method of thumb opposition that is different from the previous ones, which have some shortcomings.

As the device would cost less than 3 USD, not only does it have good sensitivity and specificity, but it is also affordable. Thus, even a primary doctor or a non-orthopedist with limited knowledge and experience in hand surgery can detect thumb opposition impairment due to diseases such as severe CTS and CM osteoarthritis. Consequently, it may be able to prevent an increase in the number of patients with progression of thenar atrophy who require opponensplasty; furthermore, in the future, this method may be used to detect suspected CTS prior to doctor consultation by using it in combination with specific questionnaires and objective evaluation apps, such as some that have been recently reported [49,50].

There are several limitations to this study. First, the evaluation was not performed on patients with mild CTS. Second, presently, the sensors can only measure the angle of single-axis movement; thus, it was not possible to measure the pronation angle of the thumb, which moves 3-dimensionally. Third, the signal of the sensor was affected by changes in dimensions. In the next step, we will plan to determine the size of the sensors based on the size of the hand. Forth, although this method used continuous variables, it was only a relative numerical evaluation method using correlation coefficients. Finally, there were some false positives and false negatives. The cases of false negatives cannot be overlooked because they may lead to irreversible atrophy. However, the sensitivity of our method for detecting opposition impairment was higher than that of previous methods (0.66, 0.19 for [47,51], respectively); moreover, in the clinical filed, we considered that this aspect could be sufficiently improved by supplementing with an easy medical interview and physical findings such as those mentioned above.

In the future, we plan to use the method to evaluate thumb opposition in patients with mild CTS, evaluate the correlation between NCV values and thumb function, and determine the cutoff values by increasing the sample size. In addition, we are aiming to play a part in the development of sensor gloves, which can evaluate not only finger flexion–extension and adduction–abduction but also thumb opposition movements.

## 5. Conclusions

We developed the thumb opposition evaluation device using CNT strain sensors and verified the reliability of the device; furthermore, we evaluated the effectiveness of the device by comparing the thumb opposition evaluation in the control group and the CTS group. The reliability of the device was excellent, and in addition, the sensitivity was higher than the existing methods. The results suggest the usefulness of this device as the detector of thumb opposition impairment. We plan to conduct further researches, including application to patients with mild CTS.

## Figures and Tables

**Figure 1 sensors-20-03998-f001:**
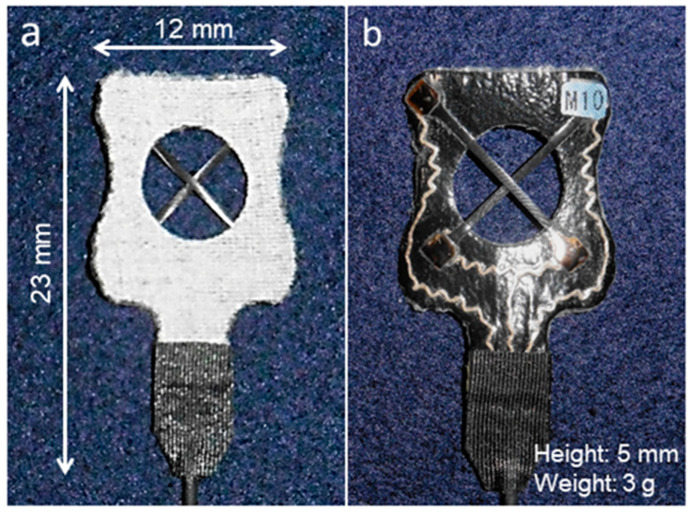
Measurement device developed in the study. Two carbon nanotube strain sensors were combined in a cross: (**a**) front and (**b**) back.

**Figure 2 sensors-20-03998-f002:**
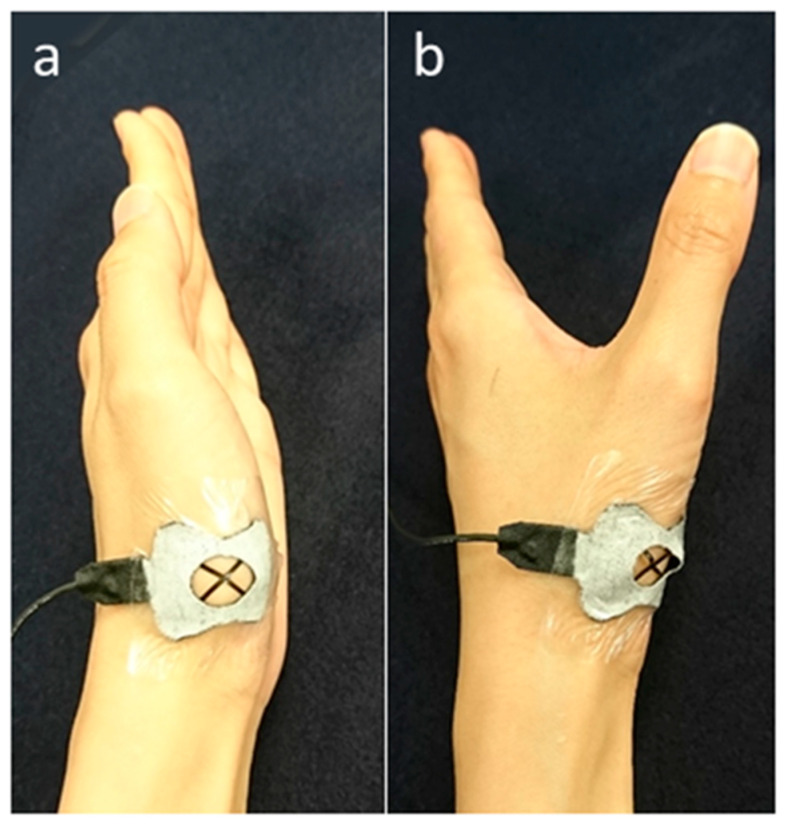
Attachment site of the sensor and position of the thumb during measurement: (**a**) adduction and (**b**) palmar abduction.

**Figure 3 sensors-20-03998-f003:**
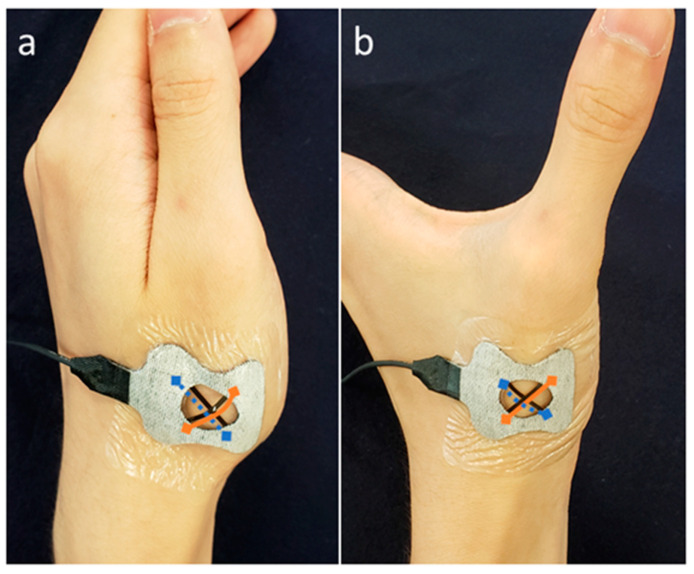
(**a**) In the adduction position, Sensor 1 (orange line) was slack and Sensor 2 (blue broken line) was taut, while to the contrary, (**b**) in the palmar abduction position, Sensor 1 was taut and Sensor 2 was slack.

**Figure 4 sensors-20-03998-f004:**
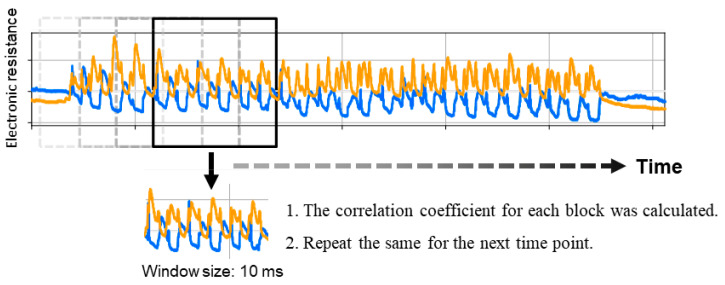
Method of calculation of the correlation coefficient transition from raw data. A part of the data was extracted as a block of 10-ms width; the correlation coefficient was calculated based on the value in the block, and this calculation was performed from beginning to end, with an interval of 5 ms.

**Figure 5 sensors-20-03998-f005:**
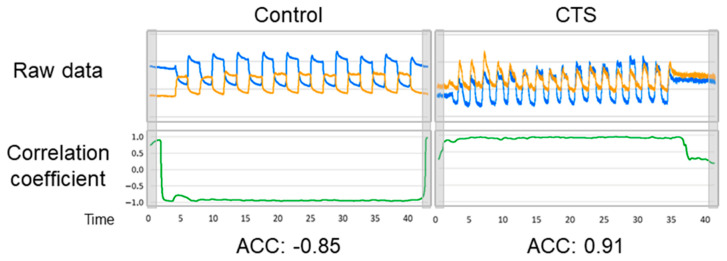
Representative examples of the correlation coefficient transition graphs in the control and carpal tunnel syndrome (CTS) groups. The average correlation coefficient (ACC) was calculated according to the above-mentioned method. The gray squares are the areas excluded in the ACC calculation.

**Figure 6 sensors-20-03998-f006:**
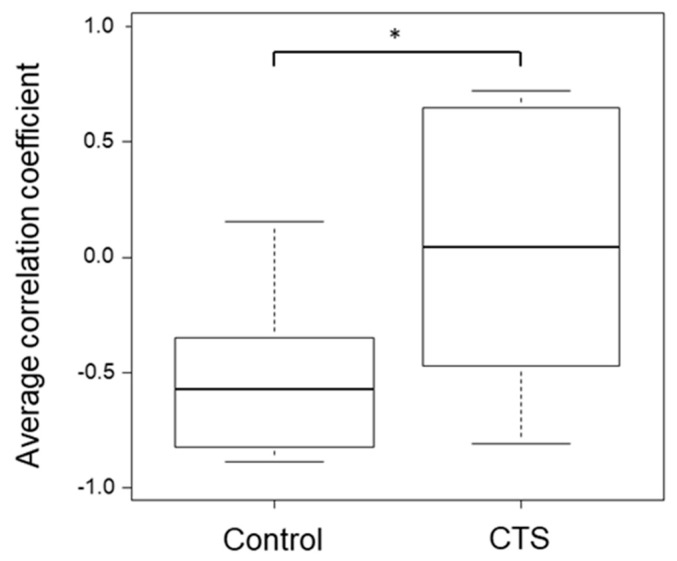
Box plot showing the average correlation coefficient in the control and carpal tunnel syndrome (CTS) group. Upper whisker, 90th percentile; box top, 75th percentile; central line, median; box bottom, 25th percentile; lower whisker, 10th percentile.

**Figure 7 sensors-20-03998-f007:**
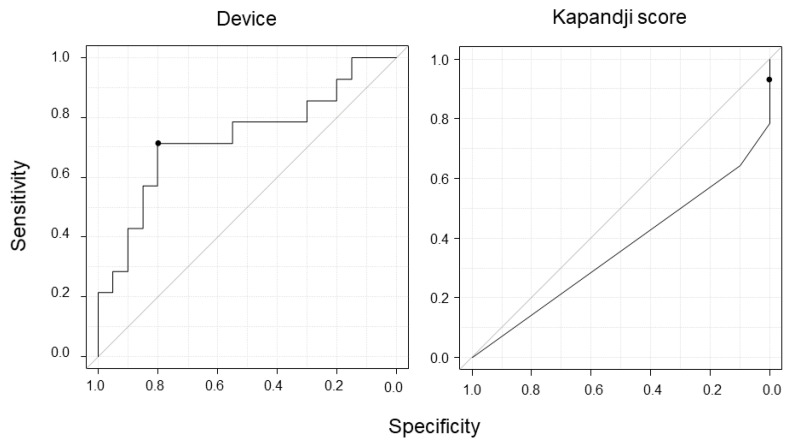
Receiver operating characteristic curves of our device method and the Kapandji score. The black dot indicates the cutoff point.

**Table 1 sensors-20-03998-t001:** Demographic features in the control and CTS groups.

	Control (n = 11)	CTS (n = 13)	*p*-Value
Age (years)	64 [58–68]	59 [55–67]	0.56
Sex (female)	10	12	
Number of hands	20	14	
Number of hands with cane	2	2	

Age is presented as median interquartile range (IQR). The number of participants, hands, and sex are presented as numerical values. Statistical significance was determined using the Mann–Whitney *U* test. CTS, carpal tunnel syndrome.

**Table 2 sensors-20-03998-t002:** Data of physical findings and NCV in the CTS group (n = 14).

Thenar Atrophy	
Absent	6
Mild	1
Moderate	3
Severe	4
Padua’s classification	
Normal	0
Minimal	0
Mild	0
Moderate	7
Severe	2
Extreme	5

NCV, nerve conduction velocity; CTS, carpal tunnel syndrome.

**Table 3 sensors-20-03998-t003:** Results of the number of hands at each Kapandji score.

Kapandji Score	Control (n = 20)	CTS (n = 14)
1	0	0
2	0	0
3	0	0
4	0	0
5	0	1 (7)
6	0	0
7	0	0
8	0	2 (14)
9	2 (10)	2 (14)
10	18 (90)	9 (65)

Data are presented as number (%). CTS, carpal tunnel syndrome.

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
