# Peer review of "Device Development for Detecting Thumb Opposition Impairment Using Carbon Nanotube-Based Strain Sensors"

_sensors, 2020, doi:10.3390/s20143998_

Round 1
Reviewer 1 Report
Ref_comments to the paper titled as “Device development for detecting thumb opposition impairment using carbon nanotube–based strain sensors@ written by the authors Tomoyuki Kuroiwa, Akimoto Nimura, Yu Takahashi, Toru Sasaki, Takafumi Koyama, Atsushi Okawa, Koji Fujita.
It is well known that after the discovery of the carbon nanotubes on 1991, so many scientists and researchers have studied these nanostructures in order to find the basic application of them. Really, due to the unique energetic levels, high value of the Young’s modulus, extended surface area, etc. features, these materials can be used in the electronics, cosmos, automobile and biomedicine area as well with good advantage.
From this point of view the paper is actual and modern.
The paper presents an excellent literary review, consisting of an analysis of more than 50 scientific and technical publications. It can be argued that the authors are well aware of the problem and quite logically found their own place among the research of other scientific teams.
It is important to notice that in these research 11 volunteers as a control group and 14 hands of 13 patients with preoperative CTS before surgery as the CTS group have been involved. Special devices based on the CNTs have been developed for making the measurements. The results were obtained for different age groups, and gender was also taken into account.
Indeed, the job is interesting and can be recommended to the express-test.
The question is the following: How long do you expect your device to work? What is its performance?
As for my local opinion, the paper can be published with minor corrections.
Author Response
Dear Reviewer 1,
We would thank you and the editor for the valuable comment which will improve the contents.
We have revised our manuscript according to your comment as follows:
- How long do you expect your device to work?
Response:
Thank you for your question.
We have added a sentence in Discussion section as follows:
“…was used for the analysis. Because this device was a wired connection, it could measure continuously for about 40 hours.” (page 3 – line 133 to 134)
- What is its performance?
Response:
We have added sentences in Materials and Methods section as follows:
“The sensor could measure its extension and contraction as an electric voltage change. It could be stretched up to 200% of the length, and could exhibit a short sensing delay of less than 15 ms. In addition, it could withstand 180,000 cycles of expansion and contraction up to 1.3 times of the length.” (page 3 – line 123 to 126)
Sincerely,
Koji Fujita, M.D., Ph.D.
Reviewer 2 Report
This is quite interesting and valuable work about the use of carbon nanotube–based strain
- Please explain the characteristics of the sensors used and the reason why these sensors were chosen.
- The device as a whole it also needs to be explained as well as the materials that were used to produce it.
- Do the dimensions of the device influence the measured signals?
- Please explain also the signals which are being presented in Figures 3 and 4. What is the difference between the yellow and blue signals? Why are the signals different as they evolve over time?
- In the data analysis procedure, page 4, section 2.5, the authors wrote “The calculation was performed using a moving average and after excluding the values of ACC at 1 s from the start and end, as the thumb sometimes has not moved yet at these time points (see details in Figures 3 and 4).”Please explain better this procedure and if possible indicate in the figures.
- Figures should be improved.
Author Response
Dear Reviewer 2,
We would thank you and the editor for the valuable comment which will improve the contents.
We have revised our manuscript according to your comment as follows:
1. Please explain the characteristics of the sensors used and the reason why these sensors were chosen.
Response:
We apologize for the lack of explanation of the sensors. Regarding the characteristics, we have revised and added a paragraph in Materials and Methods section as follows:
“We assembled a device using 2 CNT sensors (Yamaha Corporation, Shizuoka, Japan) [42]. In this sensor, millimeter-long multiwalled CNTs were unidirectionally aligned and sandwiched between elastomer layers and the elastomer was synthesized urethane resin, which exhibits low elasticity and an affinity for human skin. The sensor could measure its extension and contraction as an electric voltage change. It could be stretched up to 200% of the length and could exhibit a short sensing delay of less than 15 ms. In addition, it could withstand 180,000 cycles of expansion and contraction up to 1.3 times of the length. (page 3 – line 120 to 126)
Regarding the reason, we have added a paragraph in Discussion section as follows:
“…ACC was significantly higher and the correlation was lost.
We have previously successfully measured thumb opposition movements non-invasively using a gyroscope [24, 27]. The previous method had limitations: the device could interfere with the thumb movement and be adversely affected by skin stretching. In this study, we focused on the CNT sensor, which was thin and elastic enough to fit the skin stretching, and easy to adapt to the joint's shape.” (page8 – line 259 to 262)
2. The device as a whole it also needs to be explained as well as the materials that were used to produce it.
Response:
We have added sentences in Materials and Methods section as follows:
“We fixed the two CNT sensors (length: 17.5 mm, width: 1 mm) in an intersecting fashion at 90° to the stretch synthetic leather (thickness: 0.5 mm, Young's modulus: 8 MPa) and the sensors connected to a conducting wire which was made of silver-plated nylon fiber. Moreover, Sensors and connecting wires were placed on the leather by covering with urethane film (thickness: 0.05 mm, Young's modulus = 20 MPa) (Figure 1).” (page 3 – line 126 to 130)
3. Do the dimensions of the device influence the measured signals?
Response:
Yes, it does. We have added a limitation point in Discussion section as follows:
“…which moves 3-dimensionally. Third, the signal of the sensor was affected by changes in dimensions. In the next step, we will plan to decide the size of the sensors based on the size of the hand. Forth, although this method…” (page 9 – line 299 to 300).
4. Please explain also the signals which are being presented in Figures 3 and 4. What is the difference between the yellow and blue signals? Why are the signals different as they evolve over time?
Response:
We apologize for the lack of explanation of the signals and those colors in the figure.
Regarding the colors and difference between two signals, we have revised text, and added a sentence and a new figure with a legend (new Figure 3) in Materials and Methods section as follows:
“…between both groups. We defined the sensor from the proximal ulnar side to the distal radius as the Sensor 1 (orange line in Figure 3) and the sensor from the proximal radial side to the distal ulnar side as the Sensor 2 (blue broken line in Figure 3). (page 4 – line 169 to 171).”
“…we calculated the average correlation coefficient (ACC) of the expansions and contractions of the Sensor 1 and 2 during the motion.” (page 4 – line 172 to 173).
“Figure 3. a) In the adduction position, the Sensor 1 (orange line) was slack and Sensor 2 (blue broken line) was taut, b) while in the palmar abduction position, the Sensor 1 was taut and Sensor 2 was slack, to the contrary.”
With the addition of the new Figure 3, we have changed old Figures 3, 4, 5, and 6 to Figures 4, 5, 6, and 7, respectively. (page 5 – line 174, 176 and 210, page 6 – line 215, page 7 – line 234, 236 and 241, and page 8 – line 252).
5. In the data analysis procedure, page 4, section 2.5, the authors wrote “The calculation was performed using a moving average and after excluding the values of ACC at 1 s from the start and end, as the thumb sometimes has not moved yet at these time points (see details in Figures 3 and 4).”Please explain better this procedure and if possible indicate in the figures.
Response:
Thank you for your important pointing. To explain the procedure better, we have revised the text in Materials and Methods section as follows
“…after excluding the values of ACC at 1 s from the start and end, as the thumb was not moving in all the experiments at these time points (see the beginning and end of graphs in Figures 4 and 5).” (page 5 – line 175 to 176).
Moreover, we have modified the new Figure 5 and added sentences in the legend of the new Figure 5.
“…to the above-mentioned method. The gray squares are the areas excluded in the ACC calculation.” (page 6 – line 217).
Sincerely,
Koji Fujita, M.D., Ph.D.
Round 2
Reviewer 2 Report
The authors have improved the manuscript in accordance with the suggestions.
Therefore, I can recommended it for publication after the following minor revision:
- Please verify the references and correct reference 12, it is in capital letter.
- Please delete the titles of the figures. For example, above Figure 6 is written Figure 6. This figure is already identified in the caption.
Author Response
Dear Reviewer 2,
We would thank you for the careful reviewing and comment.
We have revised our manuscript according to your comment as follows:
1. Please verify the references and correct reference 12, it is in capital letter.
Response:
We apologize for the mistake. we have revised the text in Reference 12 as follows:
“Bunnell, S., Opposition of the thumb. JBJS 1938, 20, (2), 269-284. (page 10 – line 360)
2. Please delete the titles of the figures. For example, above Figure 6 is written Figure 6. This figure is already identified in the caption.
Response:
We have deleted all titles of figures according to your suggestion.
Sincerely,
Koji Fujita, M.D., Ph.D.